# OpenReview forum: "Model Editing as a Robust and Denoised variant of DPO: A Case Study on Toxicity"
_NeurIPS.cc/2024/Workshop/SafeGenAi — SafeGenAi Poster_

### Official Review · Reviewer_G3UF · 2024-10-08
**Interesting and solid paper**

**Rating:** 9
**Confidence:** 3

**Review:**

### Summary
The paper introduces ProFS, which is a novel model alignment method designed to enhance the safety of LLMs by reducing toxicity without requiring extensive preference data for training. It identifies and projects out harmful subspaces within model parameters, offering a sample-efficient and noise-robust alternative to DPO. The method leverages factor analysis to detect toxic subspaces, applying a projection filter that effectively mitigates toxicity during model inference. It is a technically solid paper and proposes interesting method, I strongly recommend this paper to be accepted by the workshop.

### Strengths
- ProFS demonstrates high efficiency by achieving significant toxicity reduction with considerably fewer data samples compared to DPO.
- The method shows superior robustness to noisy data, which is crucial for real-world applications where data quality can vary. This is particularly beneficial in reducing the susceptibility to mislabeled training data.
- The paper provides a strong theoretical validation attached in the appendix to elaborate its findings. As a workshop paper, it provides some interesting insights deep into the experiment design.

### Weaknesses

- There is a potential risk that by focusing too strongly on eliminating toxicity, the model might overfit to non-toxic data. Maybe more evaluation could be done to elucidate this point.

---

### Official Review · Reviewer_wccP · 2024-10-09

**Rating:** 7
**Confidence:** 4

**Review:**

**Summary**

This paper introduces ProFS (Projection Filter for Subspaces), a novel approach to reducing model toxicity by identifying and editing toxic subspaces in the activations of large language models (LLMs). The paper claims that ProFS is a more sample-efficient and robust alternative to Direct Preference Optimization (DPO), which is traditionally used for aligning models with user preferences. The work provides theoretical justification for ProFS, demonstrating its connection to factor analysis and showing how it can be viewed as a denoised version of a single DPO step. The empirical results suggest that ProFS is effective in reducing toxicity while maintaining model capabilities across multiple models and datasets.

**Strength**

- The introduction of ProFS as an editing-based alternative to tuning-based alignment methods like DPO is a valuable contribution to the field of model alignment. By leveraging factor analysis and singular value decomposition (SVD), the authors present a clear and well-motivated method for identifying and removing toxic subspaces in model activations.
- The paper highlights a key advantage of ProFS, its robustness to noisy preference data. This robustness is demonstrated both theoretically and empirically, which is particularly important in real-world scenarios where labeled preference data is often noisy or inconsistent.
- The authors conduct extensive experiments comparing ProFS with DPO across a range of models (e.g., GPT-2, Mistral) and tasks (e.g., toxicity reduction). ProFS consistently reduces toxicity with fewer samples and greater robustness to noise, providing strong empirical support for the proposed method.

**Weakness**

- The paper could be more impactful by demonstrating the effectiveness of ProFS on a wider range of tasks and model architectures. While toxicity reduction is an important challenge, expanding the scope to include other types of harmful outputs or biases would increase the broader significance of the work.

---

### Official Review · Reviewer_Xori · 2024-10-09

**Rating:** 6
**Confidence:** 3

**Review:**

The paper studies the model editing for alignment of language model in the case of toxicity. Based on the MLP structure of toxic and non-toxic subspaces, the authors propose to identify the toxic embedding, select toxic directions to get the toxic projection, and remove it from the model. The authors show the effectiveness and robustness of this method compared to fine-tuning methods in experiments.

Pros:

(1)  The authors propose a method that can identify a low-dimensional toxicity subspace of MLP and remove it from the model. This method complements the toxic identification for attention head activation in literature.

(2) The authors analyze how the proposed method identifies toxic subspaces via a factor analysis. The authors characterize the error of recovering the toxic subspace.

(3) In experiments, the authors compare the proposed method with DPO by showing improved toxicity. The authors also show better data efficiency and robustness compared with DPO.

Con:

(1) The authors assume the MLP layers have low-dimensional toxicity subspace. Although it is believed in some models, how to identify this low-dimensional structure is not investigated.

(2) The relation of the proposed MLP-based method and the attention-based toxic identification should be better explained. Can the authors provide some comparison experiments?

(3) Other than DPO, have the authors compare with other alignment methods in theory and or experiments?